# Enhancing Graph Injection Attacks Through Over-Smoothing Amplification

## Abstract

Graph Injection Attack (GIA) on Graph Neural Networks (GNNs) has attracted significant attention due to its serious threats to the deployment of GNNs by carefully injecting a few malicious nodes into graphs. Existing GIA defense methods mostly follow a framework similar to the defense depicted in the images. Instead, we aim to enhance the attack capabilities of GIA by studying the properties of the graph itself. Considering the negative impact of the over-smoothing issue in GNNs, we propose *O*ver-*S*moothing adversarial *I*njection (OSI), a plug-and-play approach that can be easily integrated with existing GIA methods to enhance the attack power by amplifying the over-smoothing issue. Specifically, OSI proposes two metrics to evaluate the over-smoothing of the graph. We prove that these two metrics are highly correlated with singular values of the adjacency matrix. Thus, we introduce a Smooth Injection Loss (SIL) to smooth the singular values. By fine-tuning the adjacency matrix using SIL, OSI can amplify over-smoothing and enhance the attack power of GIA. We conduct experiments on 4 benchmark datasets and the state-of-the-art GNNs and GIA methods. Empirical experiments demonstrate that OSI can significantly improve the attack capabilities of existing GIA methods on different defensive models.

## 1 Introduction

In recent years, Graph Neural Networks (GNNs) have seen widespread adoption for various graph-based tasks on structured relational data (Wu et al., 2020). GNNs have provided promising performance on both node-level and graph-level tasks, such as node classification (Kipf & Welling, 2016), social network analysis (Ying et al., 2018), drug discovery (Jiang et al., 2020), and computer vision (Qi et al., 2018). Despite the state-of-the-art performance, many studies have shown that GNNs are vulnerable to carefully designed adversarial attacks (Sun et al., 2018; Jin et al., 2021; Goodfellow et al., 2020), where the prediction capability can be significantly affected by small intentional perturbations on the input. The adversary can introduce new edges, delete original edges, or modify node features within a limited budget, such that the perturbations are unnoticeable and difficult for defenders to identify.

Most previous studies conduct attacks following the settings that the adversary can directly modify the edges between the original nodes and node features of the input graph, i.e., Graph Modification Attack (GMA) (Xu et al., 2019; Chang et al., 2020), as illustrated in Figure **??**. However, this setting is often impractical in the realistic scenario, since it's hard for the adversary to get access to modify the original graph. Considering the citation network, it is quite difficult for adversaries to modify existing citations or features of published papers. Therefore, Graph Injection Attack (GIA), as a relatively more practical attack setting, has drawn increasing attentions (Wang et al., 2018; Zou et al., 2021; Chen et al., 2022). In the GIA settings, the adversary can inject some malicious nodes into the graph while leaving the original edges and node features unchanged. GIA is quite flexible since the adversary can determine how to link the original nodes with the injected malicious nodes, and how to generate features for the injected nodes. Such flexibility not only brings new challenges for the adversary to devise a GIA attack strategy, but also improves the attack capability of GIA. Recent research (Chen et al., 2022) has shown that GIA can be provably more harmful than GMA due to its high flexibility. Consequently, it is significant to investigate GIA and its defense strategies.

The early GIA strategies typically conduct attacks by utilizing heuristic methods and gradient information. However, they do not consider the properties of the graph itself, e.g., the over-smoothing issue. GNNs learn node representations by iteratively aggregating representations of neighbors. Since the GNN model only leverages the information from immediate neighbors at each layer, it's straightforward to stack multiple layers when we hope to aggregate information from more distant nodes. However, deeply stacking the layers often causes dramatic degradation in the performance of GNNs. Several recent studies have claimed that this performance degradation is attributed to the over-smoothing issue (Chen et al., 2022; Li et al., 2018; Chen et al., 2020; Zhao & Akoglu, 2020). Over-smoothing makes the representations of different nodes too similar for the model to distinguish, resulting in poor performance in a variety of downstream tasks. It is worth noting that the impact of over-smoothing is compatible with the goal of GIA attacks. This observation motivates us to incorporate over-smoothing into the GIA attack.

To bridge over-smoothing with GIA, we present a universal and effective framework for GIA to improve attack power, which is called Over-Smoothing adversarial Injection (OSI)(Figure 1). OSI fine-tunes the perturbed adjacency matrix generated by GIA methods to amplify the over-smoothing issue on graphs. Specifically, we hope the model cannot distinguish the original nodes of different classes, which means the output representations should be similar for different input node features. We introduce the Feature Shift Rate (FSR) to measure the over-smoothing for original nodes. Simultaneously, for the injected malicious nodes, since we should update the features of injected nodes within a limited budget to improve the attack capability, we hope that the feature updates should be sufficiently effective to perturb the outputs. This implies that when we update input features, the changes in the output representation should not be too small, that is, to mitigate the over-smoothing on the injected nodes. We introduce the Perturbation Shift Rate (PSR) to measure the over-smoothing for injected nodes. In order to achieve these two objectives, we analyze FSR and PSR, and find that they are highly correlated with singular values of the adjacency matrix. Therefore, we further introduce the Smooth Injection Loss (SIL), which aims to smooth the singular values and integrates two objectives in an adversarial manner. We can conduct our fine-tuning strategy OSI in three steps using SIL. OSI first applies an arbitrary GIA attack to generate the perturbed adjacency matrix, then fine-tunes the adjacency matrix to minimize SIL, and finally applies the GIA attack to update the features of injected nodes.

Extensive experiments with the state-of-the-art GNN models and GIA attacks on 4 real-world datasets demonstrate that OSI can significantly improve the attack capabilities of existing GIA attacks on different defense GNN models in most scenarios. The contributions of this work are as follows:

- We introduce two metrics to evaluate the over-smoothing: FSR for the original nodes and PSR for the injected nodes, and associate the singular values with them. We elaborate on the impact of FSR and PSR on improving attack capability;

- We develop a universal fine-tuning framework Over-Smoothing adversarial Injection (OSI) that can be combined with any GIA method to improve attack power;

- We conduct experiments that demonstrate the effectiveness of OSI in promoting GIA's attack capabilities against state-of-the-art GNN models across different datasets.

## 2 PRELIMINARY

### 2.1 NOTATIONS AND PROBLEM DEFINITION

In this paper, we consider a graph $\mathcal{G} = (\mathcal{V}, \mathcal{E})$ with $|\mathcal{V}| = n$ nodes and $|\mathcal{E}| = m$ edges. The adjacency matrix of $\mathcal{G}$ is denoted as $\mathbf{A} \in \{0, 1\}^{n \times n}$, and the feature matrix of $n$ nodes is denoted as $\mathbf{X} \in \mathbb{R}^{n \times d}$. The degree matrix of $\mathbf{A}$ is denoted as $\mathbf{D} = diag(d_1, d_2, \ldots, d_n) \in \mathbb{R}^{n \times n}$, where $d_i = \sum_{v_j \in \mathcal{V}} \mathbf{A}_{ij}$ is the degree of node $i$. Each node $i$ has a corresponding label $y_i$ in the class set $Y = \{1, 2, \ldots, C\}$. We focus on the semi-supervised node classification task. That is, we train a graph neural network $f_\theta$ parameterized by $\theta$ on the training graph $\mathcal{G}_{train}$ by minimizing the cross-entropy loss $\mathcal{L}_{train}$, aiming to maximize the prediction accuracy on a set of target nodes $\mathcal{T}$ in the test graph $\mathcal{G}_{test}$.

## 2.2 Graph Convolutional Network

To address the semi-supervised node classification task on graphs, the Graph Convolutional Network (GCN) (Kipf & Welling, 2016) was proposed. GCN is usually formed by stacking multiple graph convolutional layers. The $l$-th GCN layer is formulated as:

$$\mathbf{X}^{(l)} = \sigma(\hat{\mathbf{A}}\mathbf{X}^{(l-1)}\mathbf{W}^{(l)}), \tag{1}$$

where $\hat{\mathbf{A}} = \tilde{\mathbf{D}}^{-\frac{1}{2}}\tilde{\mathbf{A}}\tilde{\mathbf{D}}^{-\frac{1}{2}}$ is the normalized adjacency matrix, $\tilde{\mathbf{A}} = \mathbf{A} + \mathbf{I}_n$ and $\mathbf{I}_n$ is the identity matrix, and $\tilde{\mathbf{D}}$ is the corresponding degree matrix of $\tilde{\mathbf{A}}$. $\sigma(\cdot)$ is an activation function, e.g., ReLU. $\mathbf{W}^{(l)}$ is a learnable weight matrix at layer $l$. $\mathbf{X}^{(l)}$ is the node hidden representation at layer $l$, and $\mathbf{X}^{(0)} = \mathbf{X}$ is the original node feature matrix.

GCN intuitively propagates each node's representation to its neighbors at each layer, then transforms the aggregated representation with a weight matrix $\mathbf{W}$ and activation function $\sigma$ to generate a new representation.

## 2.3 Graph Injection Attack

A graph adversarial attack aims to build a modified graph $\mathcal{G}'$ that minimizes the attack loss of the GNN model $f$ under the constraint $\|\mathcal{G} - \mathcal{G}'\| \leq \Delta$:

$$\begin{aligned} &\min \mathcal{L}_{atk}(f(\mathcal{G}')), \\ &\text{s.t. } \|\mathcal{G} - \mathcal{G}'\| \leq \Delta, \end{aligned} \tag{2}$$

where $\mathcal{L}_{atk}$ is the loss function during attack, typically can take $\mathcal{L}_{atk} = -\mathcal{L}_{train}$. Different from GMA, which directly modifies the adjacency matrix $\mathbf{A}$ and node features $\mathbf{X}$ of the original graph $\mathcal{G}$, GIA injects few malicious nodes within a specific budget while leaving the original structure and node features unchanged. More specifically, the modified graph $\mathcal{G}' = (\mathbf{A}', \mathbf{X}')$ constructed by GIA can be formulated as:

$$\mathbf{A}' = \begin{bmatrix} \mathbf{A} & \mathbf{E}_{atk} \\ \mathbf{E}_{atk}^{\top} & \mathbf{O}_{atk} \end{bmatrix}, \mathbf{X}' = \begin{bmatrix} \mathbf{X} \\ \mathbf{X}_{atk} \end{bmatrix}, \tag{3}$$

where $\mathbf{E}_{atk} \in \{0,1\}^{n \times n_{\mathrm{I}}}$ is the connection between the original nodes and the injected nodes, $n_{\mathrm{I}}$ is the number of injected nodes, $\mathbf{O}_{atk} \in \{0,1\}^{n_{\mathrm{I}} \times n_{\mathrm{I}}}$ is the adjacency matrix of the injected nodes, and $\mathbf{X}_{atk} \in \mathbb{R}^{n_{\mathrm{I}} \times d}$ is the features of the injected nodes. In the framework of semi-supervised node classification, the objective of GIA is to reduce the number of correct predictions of the GNN model $f_{\theta^*}$ on the target node set $\mathcal{T}$:

$$\begin{aligned} &\min_{\mathcal{G}'}|\{f_{\theta^*}(\mathcal{G}')_i = y_i, i \in \mathcal{T}\}|, \\ &\text{s.t. } n_{\mathrm{I}} \leq \Delta, deg(v)_{v \in V_{atk}} \leq b, \mathbf{X}(v)_{v \in V_{atk}} \in \mathcal{F}, \end{aligned} \tag{4}$$

where $\theta^* = \arg\min_{\theta} \mathcal{L}_{train}(f_{\theta}(\mathcal{G}_{train}))$ is the optimal weights of the model $f$ on the training graph $\mathcal{G}_{train}$, and $V_{atk}$ is the set of injected nodes. The number of injected nodes $n_{\mathrm{I}}$ is limited by a budget $\Delta \in \mathbb{Z}$, the degree of injected nodes is limited by a budget $b \in \mathbb{Z}$, and the features of injected nodes belong to set $\mathcal{F} = \{h \in \mathbb{R}^d | \min(\mathbf{X}) \leq h_i \leq \max(\mathbf{X}), \forall 1 \leq i \leq d\} \subseteq \mathbb{R}^d$, i.e., the entries of injected nodes are between the maximum and minimum entries of $\mathbf{X}$.

GIA can be generically separated into two stages. In the injection stage, the attack algorithm generates the edges between the injected nodes and the original nodes, as well as the connections among the injected nodes. Then in the optimization stage, the attack algorithm optimizes the features of injected nodes to maximize the attack effect, which usually utilizes PGD.

## 2.4 Attack Settings

In this paper, we adopt a widely used setting of GIA, that is, inductive, black-box, and evasion attack, following (Zheng et al., 2021). Inductive means test nodes are only visible during testing and invisible during training. Black-box means the adversary does not have access to the architecture, parameters, or defense mechanism of the target model. Evasion means the attack is only performed at test time, allowing the target model to train on the clean graph.

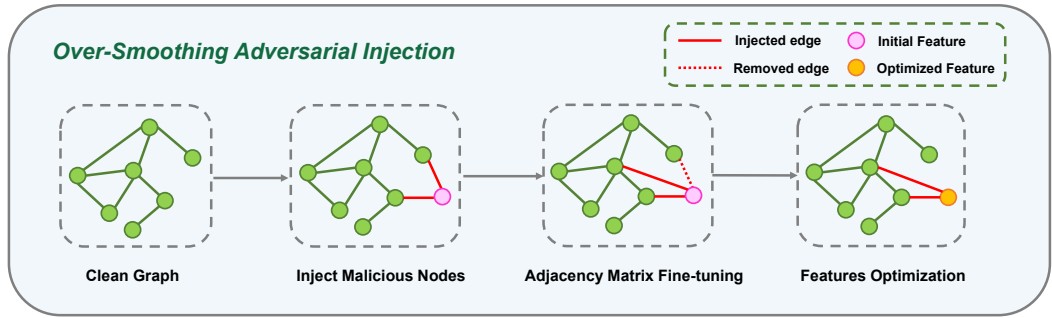

Figure 1: Overall framework of OSI

## 3 METHODOLOGY

### 3.1 OVER-SMOOTHING ISSUE IN GNNS

To obtain information about neighbors, the GNN model performs aggregation operations on node features at each layer. If the model expects to acquire information about nodes located more hops away, an intuitive method is to simply stack more GNN layers. However, as the propagation of GNN layers continues, the representations of different nodes will gradually become indistinguishable and end up reaching a stationary state(Zhao & Akoglu, 2020), i.e., the over-smoothing issue. Consider the simple GNN model SGC(Wu et al., 2019), which removes the non-linear transformation in the GCN layer. When adopting $\hat{\mathbf{A}} = \tilde{\mathbf{D}}^{r-1}\tilde{\mathbf{A}}\tilde{\mathbf{D}}^{-r}$, $\hat{\mathbf{A}}^\infty$ can be represented as(Zhang et al., 2022):

$$\hat{\mathbf{A}}_{i,j}^\infty = \frac{(d_i + 1)^r (d_j + 1)^{1-r}}{2m + n},$$ (5)

where $m$ and $n$ are the number of edges and nodes in the graph, respectively. Eq. 5 implies that when executing propagation operation infinite times, the effect of node $i$ on node $j$ is only related to their degree, which means the model cannot fully extract structure information. Consequently, the output node representation is over-smoothed and no longer applicable to downstream tasks.

These observations motivate our main idea. For the original nodes, we hope that by entering different features into the model, the output representations are as similar as possible, i.e., we should amplify the over-smoothing issue on the original nodes. Simultaneously, for the injected malicious nodes, since the features of the injected nodes will be updated iteratively, we hope that the output of the model should be sufficiently sensitive to the feature updates, i.e., we should mitigate the over-smoothing issue on the injected nodes. According to the two objectives above, we introduce two metrics to quantify the over-smoothing and devise the fine-tuning scheme OSI for the adjacency matrix to amplify the attack power of GIA. Note that the over-smoothing in OSI refers to the *feature-wise* over-smoothing mentioned in (Zhao & Akoglu, 2020). In the following, we will elaborate on our method from the perspective of the two objectives.

### 3.2 OBJECTIVE FOR ORIGINAL NODES

The GNN models perform propagation and transformation operations to capture each node's unique information and generate node representations that can be used in a variety of downstream tasks. To make the model incapable of discriminating between different nodes, we try to minimize the gap between the model's outputs for different input node features,i.e., amplify the over-smoothing issue on the original nodes. Firstly, we introduce "Feature Shift Rate (FSR)" to quantify the output modification. We formally define FSR as follows:

**Definition 3.1** (**Feature Shift Rate**). *Given a GNN model $f_\theta$, the Feature Shift Rate for the graph $\mathcal{G}$ is defined as:*

$$FSR(i,j) = \frac{Dis(f_\theta([\mathbf{X}]_i), f_\theta([\mathbf{X}]_j))}{Dis([\mathbf{X}]_i, [\mathbf{X}]_j)}, \forall 1 \le i, j \le d,$$ (6)

*where $[\mathbf{X}]_i$ denotes the $i^{th}$ column of $\mathbf{X}$, representing the $i^{th}$ dimension of node features; $Dis(\cdot, \cdot)$ is a distance metric, e.g., Euclidean distance or the inverse of cosine similarity.*

To simplify our analysis, we leverage a linearized GNN with $l$ layers, i.e., $\mathbf{X}^{(l)} = \hat{\mathbf{A}}^l \mathbf{X}$, and apply Euclidean distance as the distance metric. Thus, we have:

$$FSR(i,j) = \frac{\|\hat{\mathbf{A}}^l[\mathbf{X}]_i - \hat{\mathbf{A}}^l[\mathbf{X}]_j\|_2}{\|[\mathbf{X}]_i - [\mathbf{X}]_j\|_2}, \tag{7}$$

where $\|\cdot\|_2$ is two-norm. Our objective is to minimize FSR for the original graph so as to alleviate the distance between the output representations of the model and make them indistinguishable. With Definition 3.1, we can get the upper bound of FSR:

**Theorem 3.1.** *Given a linearized GNN $f_\theta$ trained on $\mathcal{G}$, then the upper bound of FSR for $\mathcal{G}$ is determined by $\sigma_1$:*

$$\max\{FSR(i,j), \forall 1 \leq i,j \leq d\} \leq \sigma_1^l, \tag{8}$$

*where $\sigma_1$ is the largest singular value of the normalized adjacency matrix $\hat{\mathbf{A}}$.*

The proof of Theorem 3.1 can be found in Appendix A.1. Theorem 3.1 indicates that we can minimize the upper bound of FSR by reducing $\sigma_1$. Indeed, the reduction of any singular value $\sigma_k$ can contribute to the decrease in FSR. Notably, the impact is more pronounced for larger singular values. As we hope the model generates similar representations for different inputs, i.e., $FSR(i,j)$ be small for arbitrary pairs of $i$ and $j$, we should reduce the singular values of the normalized adjacency matrix $\hat{\mathbf{A}}$. More details about FSR are elaborated in Appendix D.

### 3.3 OBJECTIVE FOR INJECTED NODES

The adversary typically conducts a GIA attack by first generating edges between the original nodes and the injected nodes, then updating the injected node features iteratively. During the update phase, we hope that the change in the model's outputs should not be too small as we update the input features, i.e., mitigate the over-smoothing issue on the injected nodes. To this end, we introduce "Perturbation Shift Rate (PSR)" to quantify the output modification in the feature updating stage. Similar to FSR, we formally define PSR as follows:

**Definition 3.2 (Perturbation Shift Rate).** *Given a GNN model $f_\theta$, the feature matrix $\mathbf{X}'^{(0)}$ denotes the features of the graph after injecting malicious nodes with initial features, and the feature matrix $\mathbf{X}'^{(k)}$ denotes the features of the graph after updating injected node features for $k$ iterations. The Perturbation Shift Rate for the injected nodes is defined as:*

$$PSR(i) = \frac{Dis(f_\theta([\mathbf{X}'^{(0)}]_i), f_\theta([\mathbf{X}'^{(k)}]_i))}{Dis([\mathbf{X}'^{(0)}]_i, [\mathbf{X}'^{(k)}]_i)}, \forall 1 \leq i \leq d, \tag{9}$$

*where $[\cdot]_i$ and $Dis(\cdot, \cdot)$ are the same as in the FSR.*

As stated in Section 3.2, we leverage a linearized GNN and Euclidean distance to instantiate PSR:

$$PSR(i) = \frac{\|\hat{\mathbf{A}}^l[\mathbf{X}'^{(0)}]_i - \hat{\mathbf{A}}^l[\mathbf{X}'^{(k)}]_i\|_2}{\|[\mathbf{X}'^{(0)}]_i - [\mathbf{X}'^{(k)}]_i\|_2}. \tag{10}$$

Since we hope to perturb the model's outputs effectively by updating the injected node features, the PSR for the injected nodes should not be too small. With Definition 3.3, we can get the lower bound of PSR:

**Theorem 3.2.** *Given a linearized GNN $f_\theta$ trained on $\mathcal{G}$, and the perturbed graph $\mathcal{G}'$, then the lower bound of PSR for the injected nodes is determined by $\sigma_s$:*

$$\min\{PSR(i), \forall 1 \leq i \leq d\} \geq \sigma_s^l, \tag{11}$$

*where $\sigma_s$ is the singular value of the normalized adjacency matrix $\hat{\mathbf{A}}'$ with the least non-zero value.*

The proof of Theorem 3.2 can be found in Appendix A.2. Theorem 3.2 indicates that the minimum of PSR is bounded by $\sigma_s$. In contrast to FSR, raising the singular values can help to increase PSR. Specifically, the smaller singular values have a stronger effect, implying that PSR is primarily influenced by tiny singular values. As we hope that $PSR(i)$ will not be too small for arbitrary $i$ and feature update perturbations, we should increase the singular values of the normalized adjacency matrix $\hat{\mathbf{A}}'$. More discussion on PSR can be found in Appendix D.

---

**Algorithm 1:** Over-Smoothing adversarial Injection (OSI)

---

**Input:** Original graph $\mathcal{G} = (\mathbf{A}, \mathbf{X})$; GIA attack strategy $\mathcal{A}$; set of target nodes $\mathcal{T}$;
**Output:** Attacked graph after fine-tuning $\mathcal{G}' = (\mathbf{A}', \mathbf{X}')$;
**Parameter:** Number of injected nodes $n_{\mathrm{I}}$; maximum degree of injected nodes $d$; fine-tuning
    epochs $K$;

1   Initialization: $\mathcal{G}' \leftarrow \mathcal{G}$; $\mathbf{E}_{atk} \leftarrow 0^{n \times n_{\mathrm{I}}}$; $\mathbf{O}_{atk} \leftarrow 0^{n_{\mathrm{I}} \times n_{\mathrm{I}}}$; $\mathbf{X}_{atk} \leftarrow \mathcal{N}(0, \sigma)^{n_{\mathrm{I}} \times d}$ ;
2   **while** $n_{\mathrm{I}} > 0$ **do**
3     Determine the batch number of injected nodes $b$ ;
4     $n_{\mathrm{I}} \leftarrow n_{\mathrm{I}} - b$ ;
      /* Step1:   Injected Edges Selection                           */
5     $\mathbf{A}' \leftarrow \mathcal{A}(\mathcal{G}')$, determine the injected edges using attack strategy $\mathcal{A}$ ;
      /* Step2:   Adjacency Matrix Fine-tuning                  */
6     **for** $k \leftarrow 1$ **to** $K$ **do**
7        $\mathbf{E}_{atk} \leftarrow$ Optimize the weight of injected edges connected to target nodes in $\mathcal{T}$ with
         Eq. 13 ;
8        $\mathbf{E}_{atk} \leftarrow$ Connect each injected nodes to $d$ target nodes in $\mathcal{T}$ with highest edge weights ;
9     **end**
      /* Step3:   Node Feature Optimization                    */
10     $\mathbf{X}_{atk} \leftarrow$ Optimize the features of injected nodes with $\mathcal{A}$ ;
11     $\mathbf{A}', \mathbf{X}' \leftarrow$ Update $\mathbf{A}'$ and $\mathbf{X}'$ with Eq. 3 ;
12     $\mathcal{G}' \leftarrow (\mathbf{A}', \mathbf{X})'$ ;
13   **end**

---

### 3.4 OVER-SMOOTHING ADVERSARIAL INJECTION

Through Section 3.2 and 3.3, we introduce two objectives in GIA attack scenarios. Specifically, the first objective is to diminish FSR by reducing the singular values, whereas the second objective is to enhance PSR by increasing the singular values. These two objectives seem to conflict with each other. However, different singular values affect FSR and PSR to varying degrees, enabling us to integrate two objectives and conduct our fine-tuning strategy Over-Smoothing Adversarial Injection (OSI) in an adversarial manner. OSI adjusts the perturbed adjacency matrix determined by GIA through Smooth Injection Loss (SIL).

**Definition 3.3 (Smooth Injection Loss).** *Given a graph $\mathcal{G}$, the Smooth Injection Loss for adjacency matrix $\mathbf{A}$ is defined as:*

$$\mathcal{L}_s = \sum_{i=1}^{k} \sigma_i - \gamma \sum_{i=k+1}^{n} \sigma_i, \tag{12}$$

*where $\sigma_i$ is the $i^{th}$ largest singular value of $\mathbf{A}$, $k$ and $\gamma$ are two hyper-parameters used to balance the effect of SIL on two objectives.*

We aim to find an optimal perturbed adjacency matrix $\mathbf{A}_s$ that minimizes the loss in Eq. 12. The first term of Eq. 12 corresponds to the large singular values that should be minimized during optimization, and the second term corresponds to the small singular values that should be maximized. Since different singular values affect FSR and PSR to varying degrees, the decrease of large singular values benefits FSR while only marginally harming PSR. Instead, increasing small singular values benefits PSR while only marginally harming FSR. Therefore, by balancing the hyper-parameters $k$ and $\gamma$, SIL can integrate two terms and identify a trade-off between two objectives, enabling SIL to benefit both.

With SIL, we can conduct our fine-tuning strategy OSI. Given a clean graph $\mathcal{G}$, consider an arbitrary GIA attack strategy $\mathcal{A}$. First, we utilize the strategy $\mathcal{A}$ to inject few malicious nodes, yielding a perturbed adjacency matrix $\mathbf{A}' = \mathcal{A}(\mathcal{G}) \in \{0,1\}^{(n+n_{\mathrm{I}}) \times (n+n_{\mathrm{I}})}$. Then in order to minimize SIL, OSI modifies the injected edges in $\mathbf{A}'$:

$$\arg\min_{\mathbf{A}'} \sum_{i=1}^{k} \sigma_i - \gamma \sum_{i=k+1}^{n+n_{\mathrm{I}}} \sigma_i. \tag{13}$$

Specifically, OSI utilizes PGD to update injected edges in $\mathbf{A}'$ in a continuous manner:

$$\mathbf{A}'_{t+1} = \mathbf{A}'_t - \eta(\mathbf{M} \odot \nabla \mathcal{L}_s(\mathbf{A}'_t)), \tag{14}$$

where $\eta$ is the learning rate, $\odot$ represents element-wise multiplication, and $\mathbf{A}'_t$ represents the adjacency matrix after $t$-th updates, i.e., $\mathbf{A}'_0 = \mathbf{A}'$. $\mathbf{M} \in \{0,1\}^{(n+n_I)\times(n+n_I)}$ is a mask matrix, which allows OSI to update only the edges connected to injected nodes without altering the edges on the original graph. Specifically, $\mathbf{M}_{i,j} = 0$ for any $0 \leq i, j < n$ and $\mathbf{M}_{i,j} = 1$ for other values. After updating the adjacency matrix with Eq. 14, OSI sets the $d$ edges with the largest weights to 1 and the other edges to 0 for each injected node, completing one epoch of updates. OSI obtains the optimal adjacency matrix by performing updates for several epochs. Note that the adjacency matrix fine-tuned by OSI should also satisfy the constraint in Eq. 4, which means OSI does not introduce more budget. OSI finally updates injected node features in accordance with $\mathcal{A}$'s update strategy, completing the GIA attack. The overall fine-tuning process of OSI is illustrated in Algorithm 1.

# 4 EXPERIMENTS

## 4.1 EXPERIMENT SETUP

**Datasets.** We conduct our experiments on 4 datasets, including two popular citation networks refined by GRB(Zheng et al., 2021), i.e., Cora(Yang et al., 2016) and Citeseer(Giles et al., 1998), and Wiki(Yang et al., 2015) from GRB. We also use large-scale network Computers(McAuley et al., 2015) to verify the scalability of OSI. For each dataset, we specify the budgets for the injected node number and the maximum degree based on the number of target nodes and the average degree of the dataset. Specifically, we inject 40 nodes into Wiki with a maximum degree of 20, 60 nodes into Cora and Citeseer with a maximum degree of 20, and 200 nodes into Computers with a maximum degree of 100.

**Evaluation protocol.** As we adopt the black-box setting in our experiments, we cannot obtain information about the target model. Thus, we should train a surrogate model to perform transfer attacks with various GIA methods. (Zou et al., 2021) demonstrates that GCN yields attacks with better transferability, which is probably because most GNN models are designed based on GCN, making them more similar to GCN. Therefore, we utilize a 3-layer GCN as the surrogate model to generate perturbed graphs. We first train the surrogate model, then perform GIA attacks fine-tuned by OSI on it, and transfer the injected nodes to all defense models. We report the prediction accuracy of defenses on target nodes to compare the attack capability of GIA methods before and after fine-tuning by OSI.

**Baseline Attack Methods.** We conduct non-targeted attacks by incorporating OSI into several existing GIA methods to verify its effectiveness and versatility. We adopt the following commonly used baseline GIA methods: (i) PGD (Madry et al., 2017); (ii) TDGIA (Zou et al., 2021); (iii) MetaGIA (Chen et al., 2022). We also incorporate HAO (Chen et al., 2022) into the GIA methods above to generate more robust attacks for homophily defenders. We provide more details about the attack methods in AppendixC.

**Baseline Defense Models.** To defend against GIA attacks, we select 5 representative GNNs as the defense models: (i) GCN (Kipf & Welling, 2016); (ii) GAT (Velickovic et al., 2017); (iii) Graph-Sage (Hamilton et al., 2017); (iv) GNNGuard (Zhang & Zitnik, 2020) and (v) RobustGCN (Zhu et al., 2019). We provide more details about the defense models in AppendixC.

## 4.2 EMPIRICAL PERFORMANCE

**Performance of OSI.** In Table 1 and 2, we report the attack performance of OSI coupled with various GIA methods on three small datasets: Cora, Citeseer, Wiki, and one large-scale dataset Computers, respectively. It is worth noting that we incorporate HAO into each GIA method to preserve the homophily distribution during the attack. For each defense model, we bold out the best attack performance and underline the second-best attack performance.

In Table 1, we can see that OSI significantly improves the attack performance of most GIA methods by a large margin of up to 33.61%. We can note that TDGIA coupled with OSI achieves most of the best performance on 3 datasets, and OSI reduces average accuracy by more than 10%. As can

Table 1: Performance of different GIA methods on 3 datasets against 5 defense models

| | | Clean | PGD | PGD | TDGIA | TDGIA | MetaGIA | MetaGIA |
|---|---|---|---|---|---|---|---|---|
| | +OSI | | | ✓ | | ✓ | | ✓ |
| Cora | GCN | 84.95 | 38.68 | 37.18 | 43.78 | **31.96** | 38.8 | 36.31 |
| | GAT | 85.69 | 42.53 | 41.29 | 43.4 | **27.86** | 41.54 | 39.55 |
| | GraphSage | 84.95 | 38.3 | 37.68 | 42.16 | **31.46** | 38.3 | 36.94 |
| | GNNGuard | 85.82 | 61.69 | 48.88 | 56.46 | **42.53** | 46.14 | 45.39 |
| | RobustGCN | 86.94 | 66.16 | 63.43 | 74.62 | 72.01 | 67.91 | **63.18** |
| | Average | 85.67 | 49.47 | 45.69 | 52.08 | **41.16** | 46.54 | 44.27 |
| Citeseer | GCN | 73.77 | 27.89 | 27.58 | 28.73 | **22.15** | 37.82 | 37.09 |
| | GAT | 73.24 | 37.93 | 36.67 | 27.37 | **21.73** | 43.26 | 42 |
| | GraphSage | 72.62 | 25.8 | **25.6** | 31.87 | 29.04 | 49.21 | 48.42 |
| | GNNGuard | 74.6 | 28.1 | **27.69** | 35 | 29.88 | 39.91 | 39.49 |
| | RobustGCN | 75.65 | 36.36 | **35.42** | 50.57 | 46.49 | 49.42 | 48.9 |
| | Average | 73.976 | 31.22 | 30.59 | 34.71 | **29.86** | 43.92 | 43.18 |
| Wiki | GCN | 74.3 | 19.72 | 17.08 | 32.36 | **12.36** | 30 | 21.94 |
| | GAT | 74.02 | 28.61 | 25.55 | 49.86 | **16.25** | 45.27 | 35.83 |
| | GraphSage | 78.47 | 40 | 32.77 | 46.25 | **32.5** | 45.41 | 37.63 |
| | GNNGuard | 72.77 | 72.63 | 72.5 | 44.3 | **25.41** | 69.16 | 65.83 |
| | RobustGCN | 75.55 | 39.16 | 34.86 | 51.66 | **27.91** | 52.77 | 47.5 |
| | Average | 75.022 | 40.02 | 36.55 | 44.89 | **22.89** | 48.52 | 41.75 |

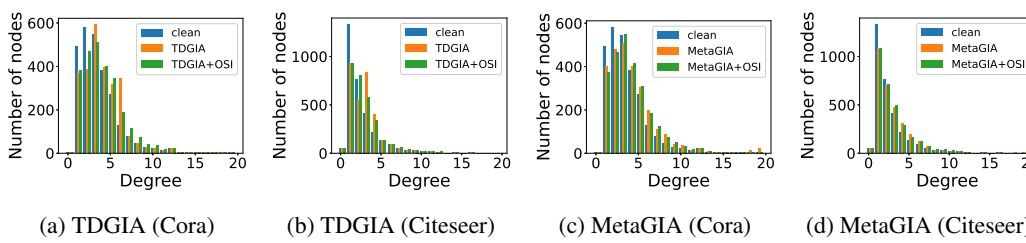

(a) TDGIA (Cora)   (b) TDGIA (Citeseer)   (c) MetaGIA (Cora)   (d) MetaGIA (Citeseer)

Figure 2: Degree distribution shift of TDGIA and MetaGIA after fine-tuning by OSI on Cora and Citeseer

be seen, almost all the best and second-best attack performances are generated by GIA methods coupled with OSI, implying that OSI is effective and transferable.

In order to verify the scalability of OSI, we also conduct experiments on large-scale dataset Computers. As shown in Table 2, OSI still improves attack performance in all scenarios, which indicates OSI can well generalize to large datasets. Particularly, TDGIA coupled with OSI achieves all the best performance in different defense models, with OSI bringing more than a 4% reduction in average accuracy. Therefore, OSI also has great scalability. To further investigate the applicability of OSI, we also conduct experiments in the transductive setting in Appendix E.

**Degree Distribution Shift.** In Figure 2, we analyze the degree distribution changes of GIA after fine-tuning by OSI on Cora and Citeseer. In real networks, degree distributions often resemble a power-law like shape. If the network after GIA attacks shows very different degree distributions, it is easy for defense models to verify and devise defense mechanisms. Figure 2a and 2b show that TDGIA will generate unreasonable degree distributions during attacks. Specifically, it generates too many nodes with degree 6 on Cora and too many nodes with degree 3 on Citeseer, which shift dramatically from the original degree distributions. While after fine-tuning by OSI, the degree distributions become more similar to the power-law distribution, which improves the unnoticeability of the attack. Figure 2c and 2d show that MetaGIA will generate some nodes with relatively high degrees, which also violates the power-law distribution. After fine-tuning by OSI, the degree distributions of MetaGIA also resemble more of a power-law shape.

Table 2: Performance of different GIA methods on Computers against 5 defense models

|  |  | Clean | PGD | PGD | TDGIA | TDGIA | MetaGIA | MetaGIA |
|---|---|---|---|---|---|---|---|---|
|  | +OSI |  |  | ✓ |  | ✓ |  | ✓ |
| Computers | GCN | 93.11 | 33.69 | 33.57 | 34.1 | **33.13** | 34.15 | 33.6 |
|  | GAT | 93.21 | 38.56 | 38.01 | 33.13 | **33.11** | 34.08 | 33.72 |
|  | GraphSage | 93.21 | 41.79 | 41.33 | 45.69 | **37.06** | 46.76 | 41.79 |
|  | GNNGuard | 91.78 | 82.1 | 81.91 | 59.46 | **55.24** | 61.3 | 61.26 |
|  | RobustGCN | 93.21 | 74.49 | 73.96 | 77.06 | **69.67** | 75.29 | 72.5 |
|  | Average | 92.9 | 54.12 | 53.75 | 49.88 | **45.64** | 50.31 | 48.57 |

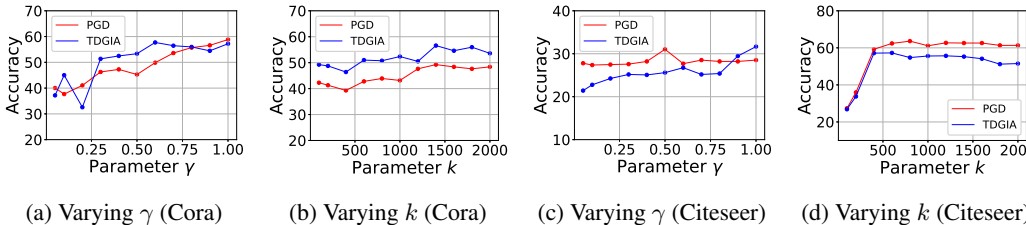

| (a) Varying $\gamma$ (Cora) | (b) Varying $k$ (Cora) | (c) Varying $\gamma$ (Citeseer) | (d) Varying $k$ (Citeseer) |

Figure 3: Attack performance of PGD and TDGIA against GCN with different parameters

### 4.3 PARAMETER ANALYSIS

As shown in Eq. 12, OSI has two critical hyper-parameters, $k$ and $\gamma$. Figure 3 shows the attack performance of PGD and TDGIA against GCN with different parameters. We can note that for different GIA methods on the same dataset, their attack performances follow a similar trend as we change parameters $\gamma$ and $k$, which indicates that OSI can efficiently transfer to other attacks when we find a set of proper parameters for an arbitrary attack. More specifically, empirical results show that the optimal values of $\gamma$ are typically around 0.1 and keep relatively stable across different datasets. The optimal values of $k$ differ for different datasets, but usually no more than one-third of the total number of nodes, which implies the second objective introduced in Section 3.3 is more crucial in OSI, i.e., mitigate the over-smoothing issue on the injected nodes.

### 4.4 LIMITATIONS

While the proposed OSI demonstrates its effectiveness and applicability, it does exhibit several limitations. Specifically, the fine-tuning process of OSI relies on computing the singular values of the adjacency matrix, which is a computationally intricate operation. This complexity becomes particularly pronounced when dealing with large-scale graphs, rendering the task of computing singular values for the entire adjacency matrix infeasible. Consequently, future research endeavors should aim to explore more efficient methods for conducting fine-tuning on large-scale graphs.

## 5 CONCLUSION

In this paper, we incorporate over-smoothing into the GIA attack and propose a general framework Over-Smoothing adversarial Injection (OSI) that can be combined with any GIA method to enhance the attack power. OSI aims to achieve two objectives by fine-tuning the adjacency matrix determined by GIA: amplify the over-smoothing issue on the original nodes and mitigate the over-smoothing issue on the injected nodes. We introduce two metrics, FSR and PSR, to quantify the level of over-smoothing. By integrating two objectives in an adversarial manner, we propose Smooth Injection Loss and conduct our fine-tuning strategy OSI. Extensive experiments with the state-of-the-art GNN models and GIA attacks on real-world datasets demonstrate that OSI can significantly improve the attack capabilities of existing GIA attacks on different defense GNN models in most scenarios. We hope that our work can provide new insights for further research on adversarial attacks on GNNs.

ETHIC STATEMENT

This paper does not raise any ethical concerns. This study does not involve any human subjects, practices to data set releases, potentially harmful insights, methodologies and applications, potential conflicts of interest and sponsorship, discrimination/bias/fairness concerns, privacy and security issues, legal compliance, and research integrity issues.

REPRODUCIBILITY STATEMENT

To make all experiments reproducible, we have listed all detailed hyper-parameters of each GIA algorithm. Due to privacy concerns, we will upload the anonymous link of source codes and instructions during the discussion phase to make it only visible to reviewers.

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

## A PROOF

### A.1 PROOF FOR THEOREM 3.1

**Theorem 3.1.** *Given a linearized GNN $f_\theta$ trained on $\mathcal{G}$, then the upper bound of FSR for $\mathcal{G}$ is determined by $\sigma_1$:*

$$\max\{FSR(i, j), \forall 1 \leq i, j \leq d\} \leq \sigma_1^l, \tag{8}$$

*where $\sigma_1$ is the largest singular value of the normalized adjacency matrix $\hat{\mathbf{A}}$.*

*Proof.* To begin with, we perform eigenvalue decomposition on the normalized adjacency matrix $\hat{\mathbf{A}}$:

$$\hat{\mathbf{A}} = Q\Lambda Q^\top = \sum_{k=1}^{s} q_k \lambda_k q_k^\top, \tag{15}$$

where $Q = [q_k]_{k=1}^s$ is the eigenvector matrix, $\Lambda = diag(\lambda_1, \cdots, \lambda_s)$ is the diagonal matrix with the elements on diagonal are the corresponding eigenvalues, and $|\lambda_i| \geq |\lambda_j|, \forall i \geq j, s \in \mathbb{N}^+$ denotes the rank of $\hat{\mathbf{A}}$. Then, we can factorize the input $x \in \mathbb{R}^n$ in the eigenspace:

$$x = \sum_{k=1}^{s} c_k q_k + \epsilon, \tag{16}$$

where $\{c_k\}_{k=1}^s$ are coordinates of $x$ in the eigenspace, and $\epsilon \in \mathbb{R}$ is a component tangent to the eigenspace. As we apply a linearized GNN, we have:

$$\begin{aligned}
\hat{\mathbf{A}}^l x &= \left(\sum_{k=1}^{s} q_k \lambda_k q_k^\top\right)^l \left(\sum_{k=1}^{s} c_k q_k + \varepsilon\right) \\
&= \sum_{k=1}^{s} c_k \lambda_k^l q_k.
\end{aligned} \tag{17}$$

Because the eigenvectors $\{q_k\}_{k=1}^s$ are mutually orthogonal, we further have:

$$\begin{aligned}
FSR(i, j) &= \sqrt{\frac{\sum_{k=1}^{s}(c_{ik} - c_{jk})^2 \lambda_k^{2l}}{\sum_{k=1}^{s}(c_{ik} - c_{jk})^2}} \\
&\leq \sqrt{\frac{(c_{i1} - c_{j1})^2}{(c_{i1} - c_{j1})^2} \cdot \lambda_1^{2l}} \\
&= |\lambda_1|^l,
\end{aligned} \tag{18}$$

follows Eq. 7, where $\{c_{ik}\}_{k=1}^s$ and $\{c_{jk}\}_{k=1}^s$ are coordinates of $[\mathbf{X}]_i$ and $[\mathbf{X}]_j$ in the eigenspace respectively. Notably, the eigenvalues and singular values of the normalized adjacency matrix $\hat{\mathbf{A}}$ differ only in sign. Hence, we can replace the eigenvalue $\lambda_1$ of Eq. 18 with the singular value $\sigma_1$:

$$FSR(i, j) \leq \sigma_1^l, \tag{19}$$

where $\sigma_1$ is the largest singular value of $\hat{\mathbf{A}}$. Thus, we complete our proof for Theorem 3.1.

$\square$

### A.2 PROOF FOR THEOREM 3.2

**Theorem 3.2.** *Given a linearized GNN $f_\theta$ trained on $\mathcal{G}$, and the perturbed graph $\mathcal{G}'$, then the lower bound of PSR for the injected nodes is determined by $\sigma_s$:*

$$\min\{PSR(i), \forall 1 \leq i \leq d\} \geq \sigma_s^l, \tag{11}$$

*where $\sigma_s$ is the singular value of the normalized adjacency matrix $\hat{\mathbf{A}}'$ with the least non-zero value.*

*Proof.* Similar to the proof of Theorem 3.1, we have:

$$PSR(i) = \sqrt{\frac{\sum_{z=1}^{s} (c_{0z} - c_{kz})^2 \lambda_z^{2l}}{\sum_{z=1}^{s} (c_{0z} - c_{kz})^2}}$$
$$\geq \sqrt{\frac{(c_{0s} - c_{ks})^2}{(c_{0s} - c_{ks})^2} \cdot \lambda_s^{2l}} \tag{20}$$
$$= |\lambda_s|^l,$$

follows Eq. 10, where $\{c_{0z}\}_{z=1}^{s}$ and $\{c_{kz}\}_{z=1}^{s}$ are coordinates of $[\mathbf{X}'(0)]_i$ and $[\mathbf{X}'(k)]_i$ in the eigenspace respectively. We also replace the eigenvalue $\lambda_s$ of Eq. 20 with the singular value $\sigma_s$:

$$PSR(i) \geq \sigma_s^l, \tag{21}$$

where $\sigma_s$ is the singular value of the normalized adjacency matrix $\hat{\mathbf{A}}'$ with the least non-zero value. Thus, we complete our proof for Theorem 3.2. □

## B  RELATED WORK

**Adversarial Attacks on GNNs.** In recent years, many studies have shown that GNNs are vulnerable to adversarial attacks, where the adversary can significantly affect the performance of GNN models by generating carefully designed perturbations on graph structure and node features. (Zügner et al., 2018) proposes the first adversarial attack on the attributed graph called Nettack. Nettack modifies both the graph structure and node features to degrade the classification accuracy of the target model. It utilizes gradient information to perform a greedy select scheme under the constraints that perturbations are unnoticeable. Metattack (Zügner & Günnemann, 2019) aims to impair the performance of the model on node classification tasks via meta-learning. It first computes the meta-gradient of the adjacency matrix to tackle a bi-level optimization problem for poisoning attacks, and then selects the perturbation with the largest meta-gradient. It can achieve significant performance reduction by only perturbing a small part of the graph. (Dai et al., 2018) proposes an attack method based on reinforcement learning, which only modifies the graph structure and can attack both the node classification and the graph classification task. (Bojchevski & Günnemann, 2019; Xu et al., 2020) propose gradient-based methods to attack graphs by adding or deleting edges. Instead of merely adding edges or deleting edges, (Ma et al., 2021) proposes ReWatt, a reinforcement learning based method that makes the perturbation unnoticeable by rewiring.

The methods above, however, need to directly modify the original graph, which is often unrealistic. Therefore, many studies focus on a more realistic scenario, i.e., graph injection attack (GIA). (Sun et al., 2020) proposes a node injection poisoning attack based on reinforcement learning. (Wang et al., 2020) proposes AFGSM under the same scenario, which utilizes an approximation strategy to linearize the model and generates perturbations effectively. (Zou et al., 2021) proposes TDGIA and presents an analysis of the topological vulnerability of GNNs under the GIA setting. TDGIA first connects injected nodes with the original nodes by the topological defective edge selection strategy, and then generates features of injected nodes by the smooth feature optimization objective. (Chen et al., 2022) finds that GIA can be more harmful than GMA under the same budget, however, GIA also causes severe damage to graph homophily distribution, which makes it easily defendable by homophily defenders. They introduce homophily unnoticeability to constrain the damage and propose HAO to instantiate it.

**Over-smoothing in GNNs.** Many previous works have shown that over-smoothing is a common phenomenon in GNNs. (Li et al., 2018) proves that the graph convolution is a kind of Laplace smoothing and the node features in a connected component will converge to similar values after repeatedly applying Laplace smoothing, and calls for paying attention to the over-smoothing issue. (Xu et al., 2018) introduces Jumping Knowledge Networks, which adopts skip connections to transmit multi-hop messages and assign different neighborhoods. (Gasteiger et al., 2019) proposes a propagation scheme based on personalized Pagerank, which can keep the locality. (Rong et al., 2020) randomly removes some edges in the input graph to mitigate message propagation, and thus mitigate over-smoothing. (Chen et al., 2020) classifies the smoothing into two kinds according to the information-to-noise ratio: reasonable smoothing that is beneficial for GNNs and over-smoothing

that causes performance reduction. (Liu et al., 2020) supposes that the crucial factor of this performance deterioration is the entanglement of representation transformation and propagation in graph convolution operations. They propose DAGNN that is able to adaptively incorporate information from large receptive fields. (Zhao & Akoglu, 2020) presents two measures for quantifying two types of over-smoothing: node-wise over-smoothing and feature-wise over-smoothing. They further propose PAIRNORM, a normalization layer for GNNs that can prevent the output features of distant nodes from being too similar, while allowing those of nodes in a neighborhood to be more similar.

## C  MORE DETAILS ABOUT ATTACK SETTINGS AND BASELINES

### C.1  ATTACK SETTINGS

**Inductive.** In the inductive setting, the training and test data are clearly separated, which means the model $f_\theta$ can only get the labeled training nodes and edges between them for training. During testing, the model can obtain the entire graph in order to predict the labels of test nodes, i.e., $\mathcal{G}_{test} = \mathcal{G}$. The node classification task could also be conducted in a transductive manner, where the model can get the whole graph that contains the test nodes without the ground-truth labels during training time, i.e., $\mathcal{G}_{train} = \mathcal{G}_{test} = \mathcal{G}$. As we conduct the evasion attack, which leads to modifications in the test graph $\mathcal{G}_{test}$, the inductive setting is more appropriate for our study.

**Black-box attack.** In the black-box setting, the adversary cannot obtain information about the target model, such as its architecture or defense strategy. However, the adversary may be allowed to obtain the original graph and labels of the training nodes to train a surrogate model. With the help of the surrogate model, the adversary can perform a GIA attack on the surrogate model, and then transfer the attack to other target models.

**Evasion attack.** We conduct the GIA attack in the evasion setting, in which the adversary only perturbs the graph during the test stage. Thus, the target models can train on the clean graph, which is different from the poison attack, where the adversary performs the attack before the training stage and the target models have to train on the poisoned graph.

### C.2  ATTACK METHODS

We conduct non-targeted attacks by incorporating OSI into several existing GIA methods to verify its effectiveness and versatility. The baseline GIA methods we adopt are introduced as follows.

**PGD(Madry et al., 2017):** PGD is a powerful adversarial attack method that has been wildly used. PGD performs attacks by utilizing local first-order information about the network. We adapt PGD to the GIA settings with black-box and evasion attacks. Specifically, we first randomly connect the injected nodes to the original target nodes, and then update the features of the injected nodes.

**TDGIA(Zou et al., 2021):** TDGIA is the state-of-the-art GIA attack, which consists of two modules: topological defective edge selection and smooth adversarial optimization. TDGIA explores the topological properties of the graph efficiently, enabling it to attack large-scale graphs.

**MetaGIA(Chen et al., 2022):** Metattack(Zügner & Günnemann, 2019) is a GMA method that uses meta-gradients to solve the bilevel problem underlying training-time attacks. Following (Chen et al., 2022), we also adapt Metattack to the GIA settings in our experiments.

**HAO(Chen et al., 2022):** HAO is a differentiable realization for regularizing the homophily distribution shift during the GIA attack. Since the baselines above may cause damage to the homophily distribution of the original graph, leading to poor performance when attacking defense models that recover the homophily. Therefore, we incorporate HAO into the GIA methods above to generate more robust attacks for homophily defenders.

We do not adopt FGSM(Goodfellow et al., 2014) and AFGSM(Wang et al., 2020), since (Zou et al., 2021) has demonstrated that TDGIA outperforms AFGSM, and FGSM has a comparable performance with AFGSM. We also exclude attack methods based on reinforcement learning, such as RL-S2V(Dai et al., 2018) and NIPS(Sun et al., 2020), since they adopt different settings from ours, and can hardly be employed in attacking large-scale graphs as well.

## C.3 Defense Models

We select 5 representative GNNs as the defense models, which are introduced as follows.

**GCN(Kipf & Welling, 2016):** GCN is one of the most frequently used baselines in graph adversarial attacks. GCN devises the convolutional architecture via a localized first-order approximation of spectral graph convolutions.

**GAT(Velickovic et al., 2017):** GAT introduces an attention-based neural network architecture to tackle node classification tasks in graph-structured data. It utilizes masked self-attentional layers to compute the hidden representations of each node and can specify arbitrary weights for different neighbors, enabling it to be applied to graph nodes having various degrees.

**GraphSage(Hamilton et al., 2017):** Instead of learning representations for each individual node, GraphSage learns a function that generates presentations by sampling and aggregating features from neighbor nodes. This function can be generalized to the nodes that are invisible during training time, enabling GraphSage to be applied to the graph that develops over time.

**GNNGuard(Zhang & Zitnik, 2020):** The core idea of GNNGuard is to detect and quantify the relationship between the graph structure and node features. With this relationship, the model assigns higher weights to edges connecting nodes with similar features, and prunes edges between dissimilar nodes.

**RobustGCN(Zhu et al., 2019):** Instead of vectors, RobustGCN adopts Gaussian distributions as the hidden representations of nodes in each layer. The model can keep robust against graph attacks by assigning different weights to node neighborhoods according to their variances in convolutions.

## D  Additional Discussions

The main idea of OSI is to fine-tune the adjacency matrix to achieve two objectives: amplify the over-smoothing issue on the original nodes and mitigate the over-smoothing issue on the injected nodes. To this end, we should reduce FSR and increase PSR simultaneously.

Appendix A.1 proofs that the upper bound of FSR for $\mathcal{G}$ is determined by the largest singular value of the normalized adjacency matrix. As shown in Eq. 18, this upper bound reaches only if

$$c_{ik} - c_{jk} = 0, \forall 2 \le k \le s,$$

which is usually unsatisfied in realistic scenarios since the distributions of coordinates $\{c_{ik}\}_{k=1}^{s}$ and $\{c_{jk}\}_{k=1}^{s}$ are uncertain. As the eigenvalues and singular values of $\hat{\mathbf{A}}$ differ only in sign, we can replace the eigenvalues of Eq. 18 with the singular value:

$$FSR(i,j) = \sqrt{\frac{\sum_{k=1}^{s} (c_{ik} - c_{jk})^2 \sigma_k^{2l}}{\sum_{k=1}^{s} (c_{ik} - c_{jk})^2}}. \tag{22}$$

Therefore, reducing any singular value $\sigma_k$ can help reduce FSR and the larger singular values have a greater effect. The analysis for PSR is similar and we can infer that the smaller singular values have a weaker effect on PSR. Observing the varying degrees to which different singular values affect FSR and PSR, we conduct our fine-tuning strategy OSI in an adversarial manner.

In this work, we only consider GIA attacks, however, the two objectives of OSI actually can be adapted to GMA settings by changing the definitions slightly. In the future, we would like to verify the effectiveness of OSI on GMA settings and involve more attack methods as well as graph tasks.

## E  More Experimental Results

### E.1  Transductive Setting

To further investigate the applicability of OSI, we also conduct experiments in the transductive setting. The experimental results are shown in Table 3. We can see that OSI can still effectively improve the attack performance of GIA methods in the transductive setting. Specifically, OSI achieves most of the best performance on 3 datasets and reduces the average accuracy by up to 11.05%, which further demonstrates the effectiveness and applicability of OSI.

Table 3: Performance of different GIA methods on 3 datasets against 5 defense models in the transductive setting

|  |  | Clean | PGD | PGD | TDGIA | TDGIA | MetaGIA | MetaGIA |
|---|---|---|---|---|---|---|---|---|
|  | +OSI |  |  | ✓ |  | ✓ |  | ✓ |
| Cora | GCN | 84.57 | 39.3 | 37.68 | 45.27 | **32.58** | 39.17 | 37.81 |
|  | GAT | 85.07 | 42.91 | 40.54 | 41.79 | **31.84** | 40.79 | 40.54 |
|  | GraphSage | 85.69 | 39.3 | 38.05 | 44.27 | **32.21** | 39.17 | 37.81 |
|  | GNNGuard | 85.94 | 60.44 | 48.5 | 57.33 | **44.77** | 47.38 | 46.76 |
|  | RobustGCN | 86.81 | **64.05** | 64.8 | 76.24 | 68.28 | 68.03 | 65.29 |
|  | Average | 85.61 | 49.2 | 45.91 | 52.98 | **41.93** | 46.9 | 45.64 |
| Citeseer | GCN | 74.5 | 27.58 | 27.27 | 28.73 | **21.42** | 33.75 | 27.89 |
|  | GAT | 72.2 | 39.6 | 27.58 | 29.25 | **26.95** | 40.64 | 39.7 |
|  | GraphSage | 71.26 | **26.12** | 26.22 | 31.97 | 27.48 | 41.58 | 37.09 |
|  | GNNGuard | 76.07 | 27.37 | 27.48 | 32.7 | **27.27** | 38.34 | 36.78 |
|  | RobustGCN | 76.28 | 34.69 | **32.49** | 52.76 | 46.08 | 49.94 | 42.42 |
|  | Average | 74.06 | 31.07 | **28.21** | 35.08 | 29.84 | 40.85 | 36.77 |
| Wiki | GCN | 73.33 | 19.3 | 17.63 | 21.38 | **16.11** | 21.8 | 17.36 |
|  | GAT | 73.05 | 19.58 | **16.94** | 44.86 | 35.97 | 28.33 | 17.36 |
|  | GraphSage | 77.63 | 39.16 | 35.13 | 46.66 | **34.16** | 42.63 | 37.08 |
|  | GNNGuard | 73.19 | 73.47 | 73.19 | 41.25 | **31.94** | 68.88 | 64.58 |
|  | RobustGCN | 75.97 | 46.94 | 46.25 | 41.66 | **37.08** | 45.83 | 44.58 |
|  | Average | 74.63 | 39.69 | 37.82 | 39.16 | **31.05** | 41.49 | 36.19 |

Table 4: Statistics of datasets

| Datasets | #Nodes | #Edges | Avg. Degree | #Features | #Classes | train/val/test | Budgets | |
|---|---|---|---|---|---|---|---|---|
|  |  |  |  |  |  |  | #Nodes | Degree |
| Wiki | 2,405 | 17,981 | 14.95 | 4,973 | 19 | 1443 / 242 / 720 | 40 | 20 |
| Cora | 2,680 | 5,148 | 3.84 | 1,433 | 7 | 1608 / 268 / 804 | 60 | 20 |
| Citeseer | 3,191 | 4,172 | 2.61 | 3,703 | 6 | 1914 / 320 / 957 | 60 | 20 |
| Computers | 13,752 | 245,861 | 35.76 | 767 | 10 | 8251 / 1376 / 4125 | 200 | 100 |

## E.2 IMPLEMENTATION DETAILS

We provide statistics for selected datasets and injection budgets in Table 4. We also provide detailed hyper-parameters for Nira in Table 5. In our experimental setup, Nira iteratively updates the adjacency matrix within a single epoch, using a learning rate $\eta$. We use a learning rate decay of 0.75 at each decay step. Nira finishes one epoch's update by normalizing the edge weights to 0 or 1. Nira completes fine-tuning after several such epochs. During the feature update phase, we perform 400 epochs of updates with a step size of 0.01.

Table 5: Detailed hyper-parameters of Nira

|  |  |  | $\gamma$ | $k$ | epoch | iteration | $\eta$ | decay step |
|---|---|---|---|---|---|---|---|---|
| Inductive | Cora | PGD | 0.05 | 200 | 3 | 20 | 2 | 5 |
|  |  | TDGIA | 0.1 | 230 | 3 | 20 | 2 | 5 |
|  |  | MetaGIA | 0.05 | 200 | 3 | 20 | 2 | 5 |
|  | Citeseer | PGD | 0.1 | 90 | 3 | 20 | 0.02 | 10 |
|  |  | TDGIA | 0.1 | 180 | 3 | 20 | 0.02 | 10 |
|  |  | MetaGIA | 0.15 | 85 | 3 | 20 | 0.02 | 7 |
|  | Wiki | PGD | 0.1 | 90 | 3 | 20 | 1 | 20 |
|  |  | TDGIA | 0.1 | 180 | 3 | 20 | 1 | 20 |
|  |  | MetaGIA | 0.1 | 120 | 3 | 20 | 1 | 20 |
|  | Computers | PGD | 0.1 | 700 | 1 | 10 | 2 | 5 |
|  |  | TDGIA | 0.1 | 700 | 1 | 6 | 2 | 6 |
|  |  | MetaGIA | 0.2 | 400 | 1 | 10 | 3 | 3 |
| Transductive | Cora | PGD | 0.1 | 210 | 3 | 20 | 2 | 5 |
|  |  | TDGIA | 0.2 | 230 | 3 | 20 | 2 | 5 |
|  |  | MetaGIA | 0.1 | 220 | 3 | 20 | 2 | 5 |
|  | Citeseer | PGD | 0.1 | 110 | 3 | 20 | 0.02 | 10 |
|  |  | TDGIA | 0.05 | 180 | 3 | 20 | 0.02 | 10 |
|  |  | MetaGIA | 0.1 | 100 | 3 | 20 | 0.02 | 7 |
|  | Wiki | PGD | 0.1 | 100 | 3 | 20 | 1 | 20 |
|  |  | TDGIA | 0.05 | 150 | 3 | 20 | 1 | 20 |
|  |  | MetaGIA | 0.1 | 100 | 3 | 20 | 1 | 20 |

