# OpenReview forum: "Enhancing Graph Injection Attacks Through Over-Smoothing Amplification"
_ICLR.cc/2024/Conference — Submitted to ICLR 2024_

### Official Review · Reviewer_ZeoF · 2023-10-20

**Soundness:** 2 fair
**Presentation:** 2 fair
**Contribution:** 2 fair
**Rating:** 3
**Confidence:** 5

**Summary:**

This paper proposes a universal attack framework to enhance the performance of Graph Injection Attack (GIA).
Specifically, the authors revisit the over-smoothing issue and introduce loss terms that encourage the occurrence of over-smoothing.
The experiment results validate the effectiveness of the proposed framework.

**Strengths:**

1. The proposed method is effective in practice.
2. The introduction of Feature Shift Rate and Perturbation Shift Rate makes sense.

**Weaknesses:**

1. The proposed FSR and PSR look similar to the homophily loss and shift loss in [1], which limits the novelty of the motivation.
2. The authors further establish the relationship between FSR, PSR, and the singular value of the normalized adjacency matrix. However, based on equation (13), it seems that all singular values are required instead of top-k.
The time complexity would be unacceptable without further optimization.
3. The practical setting with a 3-layer GCN seems to be not associated with over-smoothing, as over-smoothing occurs with very deep GCN.
For shallow GNNs, it is not a bad thing when node embeddings become similar.
How OSI degrades the performance of GNN through encouraging over-smoothing needs further discussion.
4. Minor issues: The figure referred to on page 1 is not visible.

[1] Li, Haoyang, et al. "Black-box Adversarial Attack and Defense on Graph Neural Networks." 2022 IEEE 38th International Conference on Data Engineering (ICDE). IEEE, 2022.

[2] Wu, Xinyi, et al. "A non-asymptotic analysis of oversmoothing in graph neural networks." arXiv preprint arXiv:2212.10701 (2022).

**Questions:**

1. Is every attack model in Table 1 and Table 2 enhanced by HAO? Are the ones coupled with OSI also enhanced by HAO?
2. What is the running space/time in practice? Complexity analysis is also welcomed. The Computer dataset is hard to be called large, as it only includes 13,752 nodes. Additional experiments on OGB datasets are welcome to demonstrate scalability.
3. The unnoticability issues are discussed in Figure 2, but the difference in degree distribution with/without OSI is insignificant.
I wonder if any numerical comparison is possible. Also, it would be better to include an explanation of why OSI could achieve better unnoticability.

---

### Official Review · Reviewer_Q12J · 2023-11-01

**Soundness:** 3 good
**Presentation:** 3 good
**Contribution:** 2 fair
**Rating:** 5
**Confidence:** 4

**Summary:**

The paper studies the problem of graph injection attacks (GIA) on graph neural networks. The authors propose an over-smoothing adversarial injection attack (OSI) by amplifying the over-smoothing issue to enhance the attack capability of existing GIA methods. They first introduce two terms Feature Shift Rate (FSR) and Perturbation Shift Rate (PSR) to build the connection between over-smoothing and adversarial attacks, and then propose two theorems to show that the two terms are related to the singular values of the adjacency matrix. After that, they introduce an attack method to update the inject graphs by manipulating the singular value. The experimental results on 4 public datasets demonstrate the effectiveness of the proposed method.

**Strengths:**

1. The proposed idea builds the connection between the over-smoothing and robustness
2. The paper is well-written and easy to follow.
3. The reported experimental results on 4 datasets (Wiki, Cora, Citeseer and Computers) show that OSI can enhance the attack capability of existing GIA methods.

**Weaknesses:**

1. The theorems only limited to linearized GNN models. However, in the realistic scenarios, the linearized GNNs are rarely used. Therefore, it is not clear that if in other more complex GNN models, this theorem can also hold, which limits its practical values.

2. Scalability Concerns. To execute the over-smoothing injection attack, the attacker is required to compute the singular value of the adjacency matrix during each update iteration. As pointed out by the authors, when dealing with large-scale graphs, this can lead to prohibitively long computation times. Furthermore, in their experiments, the authors claim that they utilized the large-scale graph "Computers," consisting of about 14k nodes. However, this dataset may not qualify as a genuinely large-scale dataset; it can only be categorized as a small or medium-scale dataset.

Moreover, given that OSI is designed to enhance the attack capabilities of existing GIA methods (e.g., TDGIA and AGIA), if OSI proves unsuitable for large-scale datasets (e.g., OGB-arxiv), its practical significance might be limited.

3. There are some typos on the paper. For instance, in Section 1, figure number is not correctly referenced, i.e., Figure ??

**Questions:**

Please see weaknesses

---

### Official Review · Reviewer_QKJu · 2023-11-02

**Soundness:** 2 fair
**Presentation:** 2 fair
**Contribution:** 2 fair
**Rating:** 5
**Confidence:** 4

**Summary:**

This paper focuses on graph injection attack (GAI) on graph neural networks (GNNs). Considering the negative impact of the over-smoothing issue in GNNs, this paper proposes over-smoothing adversarial injection (OSI) to enhance the attack power of existing GIA methods.

**Strengths:**

1. The studied problem is important.
2. The idea of linking over-smoothing and graph injection attacks is interesting.
3. Extensive experiments are conducted.

**Weaknesses:**

1. The complexity of the proposed method may be relatively high, since it involves calculating singular values of the adjacency matrix. More detailed complexity analysis should be better offered.
2. Lacks comprehensive discussion of existing graph injection attack methods like [1-3]. Whether the proposed method can be applied to the above methods should be experimented with.
3. The applicability of the proposed theorem and method may be limited. For example, Eq (6) may not be computable with activation functions and weight matrices. The proposed method that considers amplifying the over-smoothing issue may benefit little for target attack. Also, this paper only considers the two-stage paradigm GIA (first injection, then optimization), while it is still questionable whether the proposed method also can be applied to other one-stage paradigm GIA[1].
4. The writing of this paper needs to be further improved. For example, Line 11 in Algorithm 1, the confused use of d and b.

[1] Adversarial attacks on graph neural networks via node injections: Ahierarchical reinforcement learning approach. WWW 2020
[2] Scalable attack on graph data by injecting vicious nodes. Arxiv 2020
[3] Single node injection attack against graph neural networks. CIKM 2021

**Questions:**

1. When GNNs contain activation functions and weight matrices, whether Eq (6) is still applicable?
2. Why is FSR defined on the feature dimension instead of node pairs in Eq (6)?
3. In Appendix E, why does the proposed method in transductive setting not always improve the performance of the basic GIA method? Is there any discussion or explanation?
4. Why OSI can make the degree distributions become more similar to the power-law distribution? Is there any in-depth analysis?

---

### Official Review · Reviewer_TwVy · 2023-11-07

**Soundness:** 2 fair
**Presentation:** 2 fair
**Contribution:** 2 fair
**Rating:** 3
**Confidence:** 3

**Summary:**

This paper proposes an adversarial attack framework for graph neural networks that aim at conducting node classification tasks. The proposed framework (i.e., over-smoothing adversarial injection (OSI)) utilizes the oversmoothing issues underlying in current graph neural networks model architectures to launch adversarial attacks. Specifically, the malicious injected nodes along with their topologies promote the over-smoothing of the graph neural networks, such that node embeddings from the victim graph neural network model will be indistinguishable. The proposed method is experimented on multiple benchmark datasets and various defense/attack models with good performance downgrade to the victim model.

**Strengths:**

1. Interesting motivation from the perspective of over-smoothing. I think this is a new perspective upon understanding and interpreting existing GIAs that have been proposed for the graph machine learning community. This perspective is unique to the graph modality and deserves exploration.

2. The idea of relating FSR and PSR to singular values in the adjacency matrix is interesting as well.

**Weaknesses:**

1. Following the second strength, I think the relationship between FSR/PSR and singular values only holds for SGC according to the derivations presented in this paper. However, SGC is rarely used in the real world. The authors need to show its connection to popular backbone architectures like GCN, G-Sage, GAT, etc.

2. This weakness is also related to singular values. I think conducting SVD over large graphs (e.g., million-scale) is very computationally expensive. And if I understand this paper correctly, the proposed framework requires doing back-propagation over the whole spectrum of singular values, which is prohibitively expensive. I think this work requires more clarifications on its practical impact.

3. Following the previous weakness, experiments and computational overheads on pseudo industrial datasets should be analyzed. Examples could be OGB-Product, MAG, etc.

4. The hyper-parameter $k$ also seems very expensive to tune, which hurts the practical impacts of this paper.

5. Missing discussions on a lot of existing GIA works.

**Questions:**

Please refer to the weakness section.

---

### Official Review · Reviewer_Fsqv · 2023-11-08

**Soundness:** 2 fair
**Presentation:** 3 good
**Contribution:** 2 fair
**Rating:** 3
**Confidence:** 3

**Summary:**

The paper proposes Over-Smoothing adversarial Injection (OSI) to enhance the power of graph injection attacks. Specifically, the paper bridges over-smoothing with graph injection attacks by leveraging singular values of the adjacency matrix. They conduct extensive experiments on real-world datasets to demonstrate the effectiveness of OSI in enhancing the power of attacks.

**Strengths:**

- The introduction of over-smoothing is a fresh perspective in the graph attack domain.
- The theoretical analysis of the two metrics FSR and PSR is sufficient, which helps understand how OSI enhances attacks.
- The extensive experiment results show that OSI works well on the evaluated datasets.

**Weaknesses:**

1. This paper proposes two objectives: one is to minimize the largest singular value of the normalized adjacency matrix, while the other is to maximize the lowest singular value. Therefore, the essential objective is to decrease the difference between the non-zero singular value of the adjacency matrix. Consider an extreme case where all the non-zero singular value of the normalized adjacency matrix is the same, i.e., $\mathbf{\hat{A}} = \mathbf{U} \lambda \mathbf{I} \mathbf{V}^T$. Since the normalized adjacency matrix is symmetric (in GCN), $\mathbf{U} = \mathbf{V}$, hence $\mathbf{\hat{A}} = \lambda \mathbf{I}$. It doesn't seem to be a good idea for me to make the adjacency matrix approach an identity matrix. Can the authors explain it to me?
2. This paper adds a constraint that the output representations for the original nodes are supposed to be as similar as possible, even if those nodes belong to different classes. It may severely decrease the natural accuracy of the original nodes. An effective graph attack should maintain the performance (ACC) for those clean nodes while increasing the attack success rates. Can the authors explain how OSI can ensure natural accuracies and also give experiment results of accuracies on clean nodes?
3. The theoretical analysis is based on linear GCN and thus proposes FSR and PSR. It seems to be a strong assumption of linearity since the performance of linear GCN and nonlinear GCN is quite different.

All the above weaknesses are listed in the order of decreasing priority.

**Questions:**

1. In this paper, the authors claim that OSI can make *'the degree distribution more similar to the power-law distribution'*. This conclusion is not very easy for me to accept. I'm more inclined to regard it as a coincidence due to the dataset and the attack methods. Can the authors provide a more detailed explanation or theoretical analysis?
2. The OSI is used for untargeted attacks, i.e., decreasing the natural accuracy of target nodes. Can OSI be adaptive to targeted attacks, i.e., predicting the class of target nodes as the target class?
3. This paper does not clearly explain the relationship between injected nodes and target nodes. Is the target nodes the 1-hop neighbor of injected nodes? If so, it seems to be necessary to show that injected nodes will not affect the prediction of their 2-hop or 3-hop neighbors (clean nodes), as I mentioned in Weakness 2.


All the above questions are listed in the order of decreasing priority.

---

### Meta-Review · Area_Chair_whZQ · 2023-12-05

**Metareview:**

The paper proposes an interesting perspective on graph injection attacks by over-smoothing.

The strengths of this paper are listed below:

1. Over-smoothing and its relation to graph injection attacks is a novel perspective and worth exploring.

Weaknesses:

1. The computational cost seems to be high.

2. Lacking the analysis of non-linear GNNs, hope the authors can add it in the latter version.

Apart from the above weaknesses, there are also many other concerns addressed by different reviewers. Yet the authors don't respond to their questions. Therefore, I decided to reject this paper.

**Justification For Why Not Higher Score:**

The concerns proposed by reviewers are not addressed in the rebuttal period. And these concerns are important.

**Justification For Why Not Lower Score:**

N/A

---

### Decision · Program_Chairs · 2024-01-16

Reject